



# Past and future discharge and stream temperature at high spatial resolution in a large European basin (Loire basin, France)

Hanieh Seyedhashemi[1,2], Florentina Moatar[1], Jean-Philippe Vidal[1], and Dominique Thiéry[3]

[1]INRAE, UR RiverLy, 5 rue de la Doua CS 20244, 69625 Villeurbanne, France
[2]EA 6293 GéoHydrosystèmes COntinentaux, Université François-Rabelais de Tours, Parc de Grandmont, 37200 Tours, France
[3]BRGM, Bureau de Recherches Géologiques et Minières, BP 6009 45060 Orléans Cedex 2, France

**Correspondence:** Hanieh Seyedhashemi (hanieh.seyedhashemi@inrae.fr)

**Abstract.**

This paper presents retrospective simulations and future projections of daily time series of discharge and stream temperature for 52 278 reaches over the Loire River basin ($10^5$ km$^2$) in France, using a physical process-based thermal model coupled with a hydrological model. Retrospective simulations over the 1963–2019 are based on the Safran meteorological reanalysis over France. 21st century projections are based on a subset of the DRIAS2020 ensemble projection dataset, derived from the Euro Cordex data set through the ADAMONT statistical bias correction. Such a dataset at this large scale and high spatial resolution stands out from existing datasets, and is the first one in France derived from a physical process-based thermal model. The dataset is freely available for other studies and can be downloaded as NetCDF format from https://doi.org/10.57745/LBPGFS (Seyedhashemi et al., 2022a).

## 1 Introduction

Stream (water) temperature (Tw) is a critical parameter affecting the eutrophication of water bodies (Minaudo et al., 2018; Le Moal et al., 2019; Zhao et al., 2022), a wide range of biogeochemical processes (Ouellet et al., 2020), the life cycle (Elliott and Elliott, 2010) and spatial distribution of aquatic organisms (Cox and Rutherford, 2000; Morales-Marín et al., 2019; Picard et al., 2022). Recent evidence suggests the worldwide rise in this critical parameter due to climate change over the past decades (e.g. Moatar and Gailhard, 2006; Orr et al., 2015; Arora et al., 2016; Michel et al., 2020; Seyedhashemi et al., 2022b), which is also anticipated to continue in the future (e.g. Kwak et al., 2017; Carlson et al., 2017; Seixas et al., 2018; Du et al., 2019; Lee et al., 2020; Piotrowski et al., 2021; Michel et al., 2021). However, missing continuous long-term Tw data at a large scale over the past and future (Nelson and Palmer, 2007; Webb et al., 2008; Arora et al., 2016) has limited our understanding of large-scale controlling factors and spatio-temporal variability of thermal regimes, and of the impacts of such a variability on stream ecosystems in light of climate change (Hannah and Garner, 2015).

To overcome the lack of Tw data and to understand how the thermal regime respond to the climate change, physically-based, or deterministic, models can be used (Dugdale et al., 2017). These models simulate and project Tw dynamics through a heat budget, accounting for energy exchanges and effects of landscape characteristics on energy transfer (Sinokrot et al., 1995;





Webb and Walling, 1997; Yearsley, 2009; van Vliet et al., 2013; Beaufort et al., 2016b). Depending on the input data, these
models can be run at different temporal resolution and spatial scales, ranging from small streams to large rivers (Dugdale
et al., 2017). The outputs of these models allow detecting past and future changes in rivers' thermal regime and exploring the
influence of hydroclimatic – i.e. air temperature (Ta) and discharge (Q) – and basin drivers on such changes (see recent studies
e.g. Seyedhashemi et al., 2022b; Michel et al., 2021). For example, Seyedhashemi et al. (2022b), using a physical process-
based thermal model, found that Tw increased faster than air temperature over the past recent decades, and attributed such an
increase in Tw to the increase in Ta and decrease in Q. They also found the greatest increase in large rivers, while riparian
shading mitigated the increase in Tw in small mountainuous streams. Additionally, climate-induced changes in Tw could also
help us to predict the vulnerability of aquatic species to climate change (Lee et al., 2020).

This paper, using outputs of the T-NET physical process-based thermal model coupled with the EROS semi-distributed
hydrological model, presents daily time series of Q and Tw from the past to future at the reach scale over the Loire River
basin ($10^5$ km²), one of the largest in Europe. Section 2 briefly describes how such a dataset was produced using the above
models. Retrospective simulations based on the Safran reanalysis over the 1963–2019 period are then presented in Sec. 3.1.
Future Q and Tw projections derived from the DRIAS2020 climate projection dataset, are then presented in Sec. 3.2. Note that
retrospective simulations have also been commented previously by Seyedhashemi et al. (2022b).

## 2 Methodology for retrospective simulation and projections

### 2.1 Hydrological and thermal models

We use the 1-D Temperature-NETwork (T-NET) physical process-based thermal model coupled with the EROS semi-distributed
hydrological model to estimate daily Q and Tw for 52 278 reaches (median length=1.3 km) over the Loire River basin in France,
one of the largest in Europe ($10^5$ km²). The EROS model uses daily air temperature (Ta), precipitation (P) and potential evap-
otranspiration (PET) to simulate daily streamflow (Q) at the outlet of 368 homogeneous sub-basins (see Figure 1, top panel,
middle). Then, through a routine in the T-NET model, Q is redistributed along the river network inside each sub-basin ac-
cording to the each reach drainage area. The T-NET model uses these estimated Q, Ta, shortwave net solar radiation (Hns),
longwave radiation (Hla), specific humidity (RH), and wind velocity (W) at hourly time step to estimate hourly Tw at each
reach by solving the local heat budget, while accounting for thermal propagation and confluence mixing. Finally, hourly out-
puts of T-NET (Q and Tw) are averaged at the daily scale. Models used here do not consider the influence of water-abstractions
and impoundments i.e. simulate natural Q and Tw time series.

Detailed information on models principles, input data, calibration and validation can be found in Thiéry and Moutzopoulos
(1995); Thiéry (2018) and shortly in Seyedhashemi et al. (2022b) for EROS, and Beaufort et al. (2016a); Loicq et al. (2018);
Seyedhashemi et al. (2022b) for T-NET. For both EROS and T-NET, meteorological data (i.e. Ta, P, PET, Hns, Hla, RH
and W) for retrospective simulations (1963–2019) and projections (1976–2100) come from from different datasets described
below. Nevertheless, meterological datasets share the same 8-km grid resolution. All reaches within a grid cell are assigned
meteorological data values in that grid cell. For reaches flowing through more than one grid cell, meteorological variables



are weighted by the relative length of the reach within each grid cell (Seyedhashemi et al., 2020). Figure 1 summarises the methodology of reconstruction and projections of daily Q and Tw over the Loire River basin. Figure 2 shows the data availablity for the different simulations.

## 2.2 Retrospective simulation (1963–2019)

To reconstruct daily time series of Q and Tw over the past decades, the meteorological data provided by the 8 km gridded Safran atmospheric reanalysis data (Quintana-Segui et al., 2008; Vidal et al., 2010) released by Météo-France are used (see retrospective simulation in Figure 1). Seyedhashemi et al. (2022b) already used this set-up and resulting outputs to estimate the magnitude of past trends in Q and Tw at the seasonal and annual scales. Figure 2 shows the period concerned by this retrospective simulation. EROS had been calibrated over 1974-2018 to maximize the number of streamflow observations, with 1971–1974 used for the warm-up. The calibration optimized all unknown parameters (soil capacity, recession times, and propagation times) through maximizing the Nash-Sutcliffe Efficiency (NSE) criterion on the square root of streamflow and minimizing the overall bias (Seyedhashemi et al., 2022b, see)

## 2.3 Projections (1976–2100)

The DRIAS2020 climate projection dataset (Soubeyroux et al., 2020) has been recently released over France through the DRIAS portal (see http://www.drias-climat.fr/. It comprises an ensemble of climate projections under 3 Representative Concentration Pathways used in the fifth IPCC Assessment Report (IPCC, 2014) derived from the larger EUROCORDEX dataset using Regional Climaet Models (RCM) over Europe. This ensemble id downscaled over France to the 8-km Safran grid and bias-corrected with respect to the Safran reanalysis with the ADAMONT method (Verfaillie et al., 2017). In this study a subset of 3 contrasted future projections (GCM+RCM) are used to sample the dispersion of the full ensemble of 12 GCM+RCM projections from the DRIA2020 dataset. The 3 selected projections include a warm and wet couple of models (IPSL-CM5A/MRWRF381P), an intermediate one (CNRM-CM5-LR/ALADIN63), and a hot and dry couple (HadGEM2/CCLM4-8-17).

All these three GCM/RCMs include RCP 4.5 and 8.5, which are intermediate and extreme scenarios corresponding to a plausible representation of the future behavior of human societies. The CNRM-CM5-LR/ALADIN63 model also includes RCP 2.6. Therefore, the projections are conducted under 7 projections in total (total= (2 GCM/RCMs × 2 RCPs ) + (1 GCM/RCM × 3 RCP)= 7 projections). For each GCM/RCMs, there are two periods of data: 1) the period with GCMs forced by historical concentrations in greenhouse gases between 1976 and 2005, and 2) the projection part using RCPs as forcings, which extends from 2005 to 2100 (see Table 1, and projections in Figure 1 and Figure 2). It should be noted that, although selected GCM/RCMs have data from the 1950s, both hydrological and thermal models in the current study are used from the 1970s, onwards.

The meteorological variables provided by these 7 projections are integrated into both the EROS and T-NET models as explained in section 2.1 (see also Figure 1). Both the EROS and T-NET models are run under present land cover/land use while calibrated parameters of EROS are kept as for the retrospective simulation (see Seyedhashemi et al. (2022b) for short description of calibration). The EROS model is first executed to produce daily Q under each projection over the historical period (1976–2005) and the future (2005–2100). Indeed, for each projection, the EROS model is run from the 1976 to the 2100



**Figure 1.** Synthetic diagram showing the methodology of retrospective simulation and projections of daily Q and Tw over the Loire River basin. The top panel shows the spatial resolution of input data in each step. The red circle points show the outlets of 368 sub-basins. The three black triangle points show the position of sub-basin examples in the southern (L'Allier at Monistrol-d'Allier), middle (L'Arnon at Méreau [Pont de Méreau]) and northern part (La Loire at Montjean) of the basin used in the text. Solid black lines show the three main Hydro-Ecoregion (HER) delineations in the basin identified by Wasson et al. (2002) through grouping homogeneous areas in terms of land use/land cover, geology, and climate conditions (see Figure 1 of Seyedhashemi et al., 2022b).

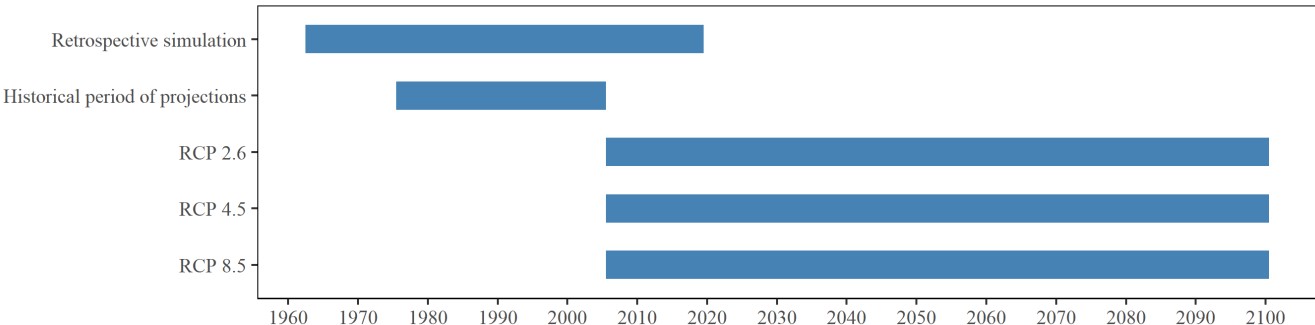

**Figure 2.** Period of available daily Q and Tw data for retrospective simulation and projections.

**Table 1.** The GCM/RCMs and RCPs used in current study study. More information can be found in http://www.drias-climat.fr/.

| GCM | RCM | HISTO | RCP 2.6 | RCP 4.5 | RCP 8.5 | Data period |
|---|---|---|---|---|---|---|
| **IPSL-CM5-MIR** | **WRF381P** | ✓ | | ✓ | ✓ | 1976-2005; 2006-2100 |
| Dufresne et al. (2013) | Skamarock et al. (2008) | | | | | |
| Hourdin et al. (2013) | | | | | | |
| **CNRM-CM5** | **ALADIN63 V2** | ✓ | ✓ | ✓ | ✓ | 1976-2005; 2006-2100 |
| Voldoire et al. (2013) | Colin et al. (2010) | | | | | |
| | Bador et al. (2017) | | | | | |
| **HadGEM2-ES** | **CCLM4-8-17** | ✓ | | ✓ | ✓ | 1976-2005; 2006-2099 |
| Jones et al. (2011) | Keuler et al. (2016) | | | | | |

while for RCPs of each GCM/RCM, the historical part (1976–2005) is the same (see projections in Figure 1, and Figure 2).
Then, future daily Q and meteorological variables provided by GCM/RCMs are used in the T-NET thermal model to produce
daily Tw under these future climate projections over the whole century (1976–2100). It should be noted that like projected Q,
for RCPs of each GCM/RCM, projected Tw over the historical part (1976–2005) is the same (see projections in Figure 1, and
see Figure 2).

**3   Data assessment and overall description**

Although the time series of both Q and Tw are available for the whole year, here, we mainly focus on June to August months
(hereafter referred to as summer), the time of the year which is vital for the survival (Steel et al., 2017), growth and migration
of aquatic communities (Arevalo et al., 2020).





## 3.1 Retrospective simulation (1963–2019)

**Streamflow**

The performance of the EROS model in simulating daily Q over the Loire River basin was assessed in a previous paper
(Seyedhashemi et al., 2022b). In a majority of calibration stations (75%), and stations on the French Reference Hydrometric
Network (83%), the Nash-Sutcliffe efficiency of simulated daily Q is $> 0.7$ for Q, $ln(Q)$, and $\sqrt{Q}$. At the seasonal scale and
at natural calibration stations over the Loire River basin, the EROS model performed well at the annual scale while it slightly
underestimated winter and spring Q, and overestimated summer and autumn Q (see Figure S6 Seyedhashemi et al., 2022b).
Such an overestimation in Q over summer and autumn is attributed to the fact that EROS does not consider the influence of
water abstractions and impoundments (see section 2.1). Significant correlations between seasonal Q trends in retrospective
simulation against observations at hydrometric stations with long-term continuous daily data were noted (see Figure S10
Seyedhashemi et al., 2022b).

Lastly, decreasing Q trends were detected in the southern part of the basin (in the Massif central) over the 1963–2019
period (up to -16 %/decade) while mainly increasing Q trends was found in the remaining parts of the basin (see Figure 3
Seyedhashemi et al., 2022b). Figure 3, left panel, shows such a decrease in summer Q for a sub-basin in the southern part
while summer Q is relatively stationary at the other two sub-basins in the middle and northern part. It should be noted EROS
performed well in reconstructing daily Q at the outlet of these 3 sub-basins (see Figure S8 Seyedhashemi et al., 2022b).

**Stream temperature**

A small underestimation in seasonal Tw on large rivers was found in a previous paper (see Figure S9 Seyedhashemi et al.,
2022b). Indeed, 3 %–83 % stations (resp. 50 %–100 %) on small and medium (resp. large) rivers had a RMSE $< 1 °C$ across
seasons (see their Figure S9, bottom panel). At the seasonal and annual scales, a strong coherence and agreement between
observations and reconstruction was also found for the four stations along the main stem of the Loire River with the long-term
data (see Figure 2 Seyedhashemi et al., 2022b). A significant correlation between seasonal and annual Tw trends in retrospective
simulations against observations were also found at Tw stations with long-term continuous daily data (see Figure S11 of
Seyedhashemi et al., 2022b).

A Tw increase was also detected for almost all reaches in all seasons (mean=+0.38 °C/decade) over the 1963–2019. Indeed,
the median summer Tw over the basin has increased by +0.44 °C/decade (i.e. +2.5 °C over the whole 1963–2019 period)
(according to Seyedhashemi et al., 2022b). Such a consistent increase in summer Tw in retrospective simulations can be also
seen in Figure 4, top panel. Nevertheless, the mean summer Tw in the retrospective simulation remains rather low ($< 18 °C$)
across the basin and only 14 % of reaches in retrospective simulation have a summer Tw $> 18 °C$ (Figure 5).

Figure 6, left panel shows a pretty good performance of the T-NET model in reconstructing daily Tw at the Avoine on the
Loire River (uninfluenced by human impacts) in 2003, the hottest year in the recent period (Moatar and Gailhard, 2006; Bustillo
et al., 2014; Seyedhashemi et al., 2022b). Although there is a good coherence between simulations and observations over the
year (Bias=0.7 °C), an overestimation in simulation (2.5 °C) is observed at the day with maximum daily Tw. Beaufort et al.

**Figure 3.** Summer Q in retrospective simulation and projections under RCP 8.5 for 3 sub-basins in the southern (L'Allier at Monistrol-d'Allier), middle (L'Arnon at Méreau [Pont de Méreau]) and northern part (La Loire at Montjean) of the Loire basin as shown in Figure 1, top panel. The dashed line represents the average of summer Q in the retrospective simulation over the 1963–2019 period. Blue lines show a locally weighted smoothing of the annual time series (span = 0.75).

(2016a) showed that the Root Mean Square Error (RMSE) of the T-NET model in simulating daily Tw was on average 1.60 °C at 128 natural stations with missing years over the 2008–2012 period over the Loire River basin. Over this basin, the RMSE of the T-NET model in simulating daily Tw at 275 natural stations with missing years over the 2008–2018 period (spotted through 135 "thermal signatures" approach in Seyedhashemi et al., 2020) is 1.80 °C.

In 2003, the majority of reaches (76 %) had a maximum daily Tw > 22 °C, and 49% of reaches showed maximum daily Tw > 24 °C (Figure 6, right panel). The maximum observed daily Tw in the downstream part of the Loire river at Avoine is

**Figure 4.** Time series of summer Tw in retrospective simulation and projections. The solid line and shaded area represent the median and the 10th-90th percentile band over all 52 278 reaches, respectively. The dashed lines show the average of the median summer Tw values (solid line) in the retrospective simulation over the 1963–2019 period.

31 °C in 2003 (see Figure 6, left panel). Such a value is expected to be seen at rivers with low velocity and shallow water such as Avoine is located on a large river (SO=8). However, all of 470 reaches with at least one day with Tw > 31 °C in 2003 are not located on large rivers (see Figure 7). 57 % of such reaches have a Strahler order between 5 and 6, and 12 % have a Strahler order less than 5 (see Figure 7), indicating an overestimation maximum daily Tw.





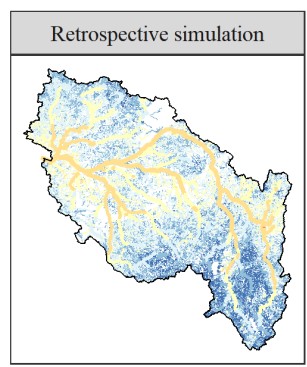

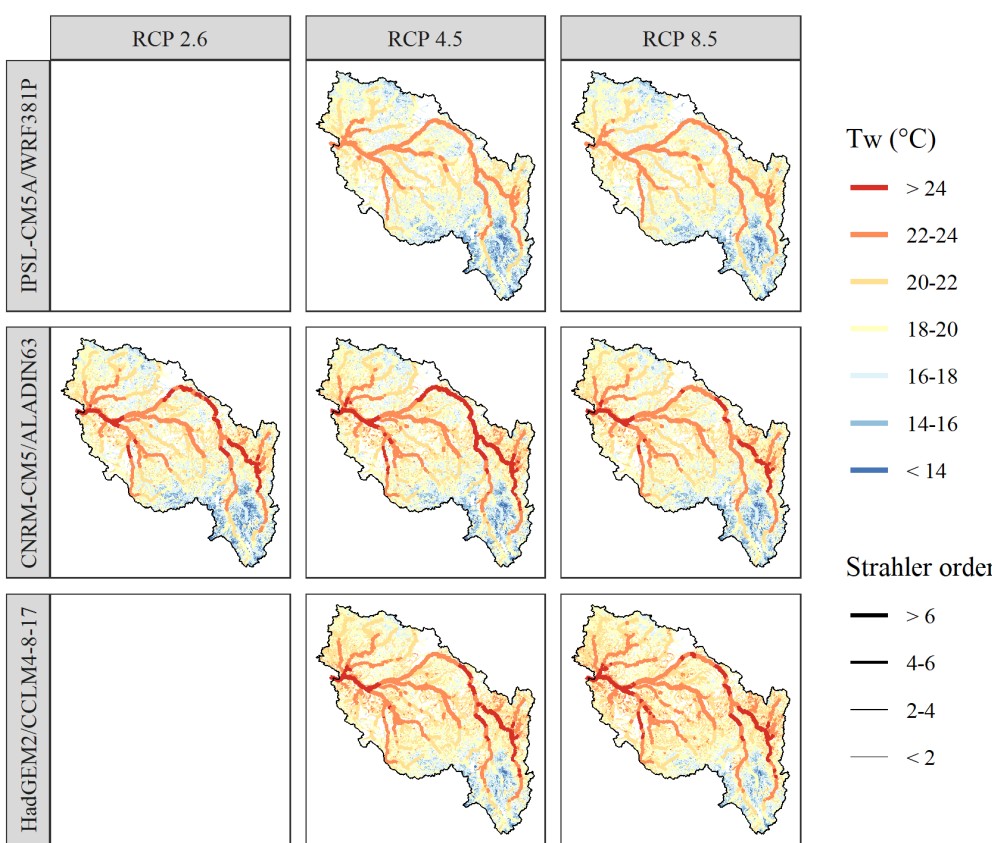

**Figure 5.** Spatial variability of summer Tw in the retrospective simulation over the 1963–2019 period and in projections for all GCM/RCMs under RCP 8.5 in the middle of century (2040–2069). Figure S2 presents the same spatial variability at the end of the century (2070–2099).

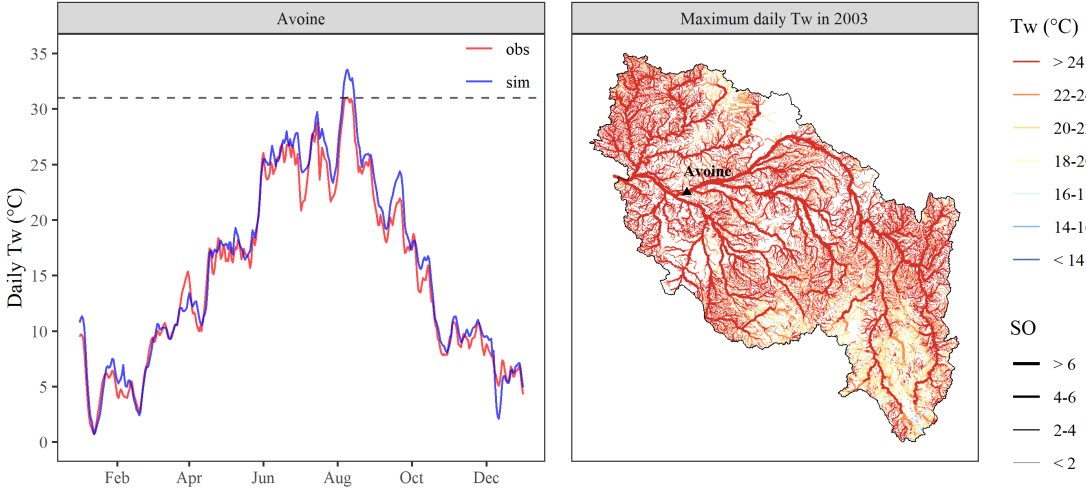

**Figure 6.** (left) Observed and simulated daily Tw at Avoine on the downstream part of the Loire River in 2003. The dashed line shows the maximum daily Tw over observed data at Avoine. (right) Simulated map of maximum daily Tw in 2003. The black triangle shows the position of Avoine on the Loire River. The line size shows the Strahler order of REACHED.

## 3.2 Projections (1976–2100)

**Streamflow**

In the southern (L'Allier at Monistrol-d'Allier) and northern part (La Loire at Montjean) of the basin, summer Q is decreasing in projections regardless of the GCM/RCM, with the largest decrease under the HadGEM2/CCLM4-8-17 model combination. However, such a decrease is limited in middle part of the basin (L'Arnon at Méreau [Pont de Méreau]) (Figure 3). Under the IPSL-CM5A/MRWRF381P model, there is a north-to-south and increase-to-decrease gradient in the middle of the century (2040–2069) with respect to the present time (1990–2019) (Figure 8). There is also a decrease in the downstream part of the basin under the HadGEM2/CCLM4-8-17 model and to a lesser extent under the CNRM-CM5-LR/ALADIN63 model. However, under the later, increase in summer Q is observed in some parts in the south while under the HadGEM2/CCLM4-8-17 model, a decrease in summer Q is projected for the whole basin with the greatest decrease in the southern part of the basin (Figure 8).

Under RCP 8.5, the annual regime of projected Q will be also different from one GCM/RCM to another, and from one sub-basin to another (see Figure S1). For instance, at a sub-basin in the southern part of the basin (L'Allier at Monistrol-d'Allier), the highest Q is projected under the HadGEM2/CCLM4-8-17 model over spring while this happens over winter period under the IPSL-CM5A/MRWRF381P model for a sub-basin in the northern part of the basin (La Loire at Montjean). Nevertheless, for both sub-basins, the annual regime of Q under the HadGEM2/CCLM4-8-17 model and RCP 8.5 shows that the lowflow period lasts longer (even until fall) compared to other two models.



Earth System
Science
Data

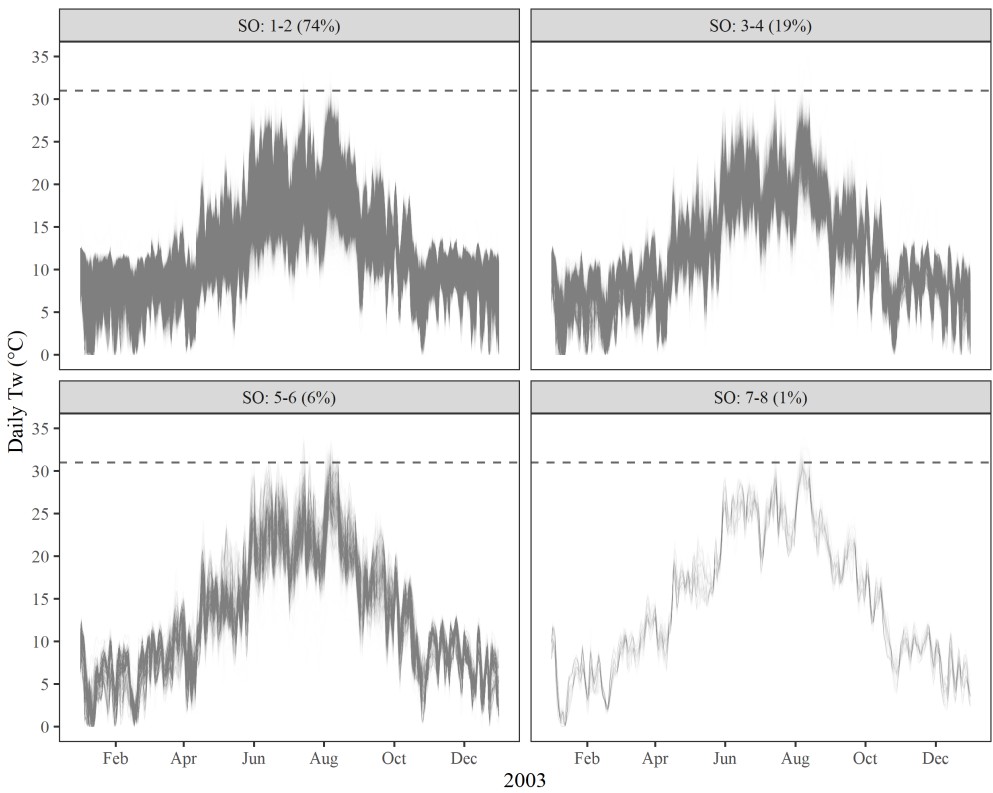

**Figure 7.** Simulated Tw for reaches with different Strahler orders (SO) in 2003. Each curve corresponds to Tw time series of one of the 52 278 reaches in the basin. The dashed line shows the maximum observed daily Tw at Avoine on the Loire River (see Figure 6) Figure 6, left panel is indeed an example of daily Tw time series shown in this figure, bottom right panel for OS $\geq$ 7. Panel titles give the percentage of reaches within each SO class.

### Stream temperature

Time series of all reaches under all GCM/RCMs show a consistent increase in summer Tw from the past to future under RCP 8.5 (see Figure 4). Under this RCP, summer anomalies with respect to 1963-2019 ranges between 5.8 °C and 7.8 °C depending on GCM/RCMs, in average over the basin at the end of the century. Conversely, summer Tw under RCP 2.6 and 4.5 are more stable after 2050 (Figure 4). Nevertheless, under these two RCPs, anomalies from 2050 onwards are yet quite large (4.2 to 4.7 °C depending on GCM/RCMs and RCP). These overall conclusions are exemplified in Figure S3.

Figure 5 shows a considerable increase in mean summer Tw in the middle of the century (2040–2069) compared to the retrospective simulation over the 1963–2019 period. Only 14 % of reaches had a mean summer Tw > 18 °C while in the middle of century, 42-73 % of reaches exhibit a mean summer Tw > 18 °C depending on the GCM/RCM and RCP. Indeed, the frequency of reaches with Tw > 18 °C is increasing (57-96 % of reaches) towards the end of the century, with the exception of the IPSL-CM5A/MRWRF381P model under RCP 4.5 and the CNRM-CM5-LR/ALADIN63 model under RCP 2.6 (see Figure S2). For



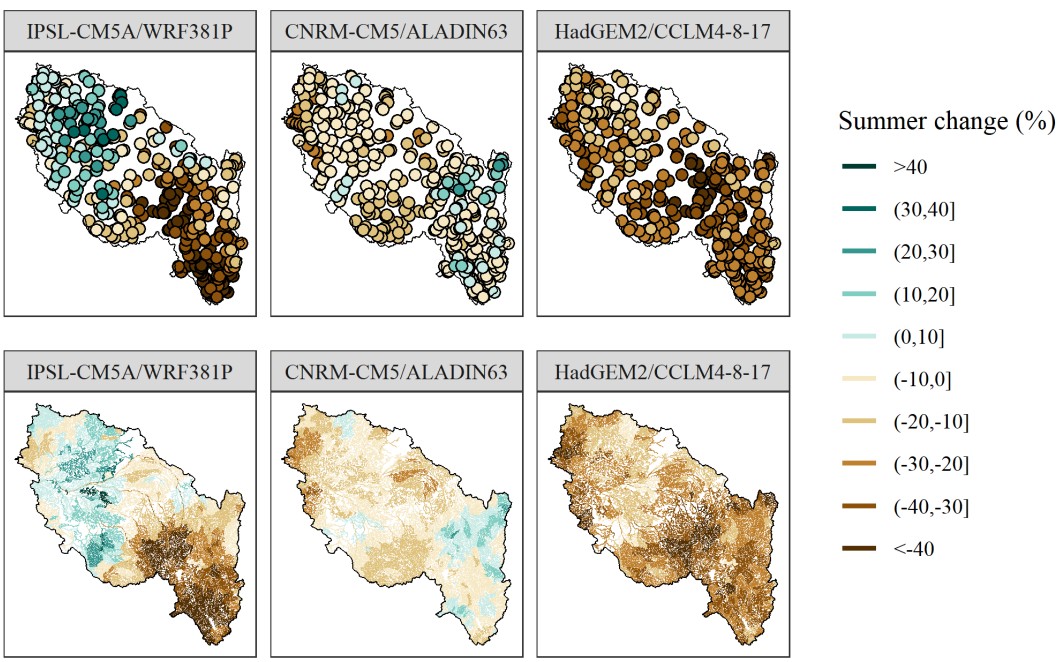

**Figure 8.** Changes in summer Q with respect to the 1990–2019 period in the middle of the century (2040–2069) for all GCM/RCMs under RCP 8.5 (top) at the outlet of 368 sub-basins and (bottom) at the reach scale.

the three selected sub-basins, an increase in the frequency of days Tw > 25 °C is also found towards the end of the century regardless of GCM/RCM under RCP 8.5 with the largest values at the end of the century ( > 50 days; see Figure S4).

## 4 Caveats

The Nash-Sutcliffe efficiency of reconstructed daily Q by the hydrological model was pretty good and > 0.7 for Q, $ln(Q)$, and $\sqrt{Q}$ (Seyedhashemi et al., 2022b). Nevertheless, there was an overestimation in summer and fall Q (Seyedhashemi et al., 2022b) as the EROS hydrological model does not consider the influence of water-abstractions and impoundments. Therefore, the users should be careful with this on highly regulated rivers. On the other hand, The RMSE of the thermal model in simulating daily Tw at 275 natural stations with missing years over the 2008–2018 period was 1.80 °C. An overestimation in reconstructed maximum daily Tw was also found at Avoine (2.5 °C). However, at the seasonal scale, no systematic bias was found for Tw at the stations located on small and medium rivers, while there was a small underestimation in seasonal Tw on large rivers (see Seyedhashemi et al., 2022b).





# 5 Conclusion

This data paper presented and described daily Q and Tw reconstructions over the 1963–2019 period as well as projections over the 1976–2100 period for 52 278 reaches over the Loire River basin ($10^5$ km$^2$) using a physical process-based T-NET thermal model coupled with the EROS hydrological model.

Daily Q and Tw are projected under three contrasted downscaled and bias-corrected climate projections (GCM/RCM) including warm and wet (IPSL-CM5A/MRWRF381P), intermediate (CNRM-CM5-LR/ALADIN63), and hot and dry (HadGEM2/CCLM4-8-17) models from the DRIAS-2020 dataset (Soubeyroux et al., 2020), under three Representative Concentration Pathways (RCPs) from the fifth report of IPCC (IPCC, 2014). All of these three GCM/RCMs were run under RCP4.5 and RCP8.5, and the CNRM-CM5-LR/ALADIN63 model was also run under RCP 2.6.

The potential applications of the proposed dataset over the past and future are manifold. This can be employed to understand spatio-temporal variability in Q and Tw, to assess the synchronicity of extermes (e.g. studies of Arismendi et al., 2013; Arevalo et al., 2020), to better explain and predict the possible spatial distribution of aquatic communities (e.g. study of Picard et al., 2022, who used the current dataset), and to assess the various stresses on freshwater habitat due to climate change (e.g. Lee et al., 2020).

# 6 Data availability

The daily Q and Tw in retrospective simulation over the 1963–2019 period and under 7 projections over the 1976–2100 period are available for T-NET hyrographic network (52 278 reaches) under the Attribution-NonCommercial 4.0 International (CC BY-NC 4.0) in NetCDF file format through: https://doi.org/10.57745/LBPGFS (Seyedhashemi et al., 2022a). Ta and other desired meteorological variables corresponding to each reach can be extracted from the closest grid cell to the reach of the Safran reanalysis database (available upon request from Météo-France) for the retrospective simulation and of the DRIAS-2020 dataset for projections (see DRIAS: http://www.drias-climat.fr/, portail partenarial Météo-France, IPSL, Cerfacs, and Soubeyroux et al., 2020).

*Author contributions.*  HS developed the dataset and prepared the manuscript. DT ran the EROS model and provided discharge data for both past and future. All co-authors contributed to the manuscript.

*Competing interests.*  The authors declare that they have no conflict of interest.

*Acknowledgements.*  The authors would like to thank Météo-France for providing the Safran reanalysis data. This work was performed in the course of a doctoral project at the University of Tours, funded by the European Regional Development Fund (Fonds Européen de



développement Régional-FEDER), POI FEDER Loire (grant no. 2017-EX001784), Le plan Loire grandeur nature, AELB (Agence de l'eau Loire-Bretagne), INRAE (l'Institut national de recherche pour l'agriculture, l'alimentation et l'environnement), and EDF (Hynes team).



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
