# Peer review of "Past and future discharge and stream temperature at high spatial resolution in a large European basin (Loire basin, France)"

_Earth System Science Data, 2022_

## Author Comment (AC1)

We thank the reviewer for his very useful comments. By addressing these comments, we believe the paper will be significantly improved, particularly with respect to a refined focus and a more detailed description of hydrological and thermal models. The reviewer comments are in italics, our responses are in normal font, and the proposed text additions and modifications are in bold.

Please note that in the following, "P", "L" and "SM" stand for page number, line number and Supplementary Materials, respectively.

**#Reviewer 1:**

*This manuscript confuses me slightly, as it is halway between a dataset description and an article analyzing trends in streamflow. I would rather this this document stripped of all of its trend descriptions and analysis and simply, and dryly, present the dataset. I do see value in the dataset itelf, but the accompanying manuscript needs a fair bit of work in my opinion.  Principly, the authors must go into much more detail about all the 'off the shelf' models they have used instead of relying on citations, and they must also qualify many of their statements and choices- the statistical tests here are not rigorous and the language is imprecise in communicating about the data. The manuscript is also oddly organized, with information relevant to methods appearing in multiple sections (see below). I recommend a major revision instead of a reject only because the dataset itself could be useful to archive. If the authors wish to do so, I beleive they must redo this accompanying document. My comments below are not exhaustive- there are many similar instances to what is below that must be found and elminated/changed.*

We would like to thank the reviewer for the thoughtful assessment. We will rewrite and reorganize the paper to better present each of models principles, input data, performances  as well as datasets. We will also add projections biases in the present-day period, which we did not include in the initial version. In this regard, we will modify the paper outline as follows:

1. Introduction
2. Models, data, calibration and validation
   2.1 Semi-distributed hydrological model (EROS)
       2.1.1    Principles
       2.1.2    Input data (1963 – 2019)
       2.1.3    Calibration and validation
       2.1.4    Future scenarios (1976-2100) and biases (1976-2005)
   2.2 Thermal physical process-based model (T-NET)
       2.2.1    Principles
       2.2.2    Input data (1963-2019)
       2.2.3    Validation
       2.2.4    Future scenarios (1976-2100) and biases (1976-2005)
3. Results
   3.1 Daily retrospective simulation (1963-2019)
       Past discharge
       Past stream temperature
   3.2 Daily projections under different future climate models and scenarios (1976-2100)
       Future discharge

Future stream temperature

4. Conclusion

*L51- For a dataset description paper, I don't think you can rely on these citations. I recognize that this paper does not describe those models, but it is not sufficient to simply list 'principles, inputs, calibration, and validation' without proof of the skill of the model nor its methods.*

We agree. We believe the response to the previous comment will address this one.

The models performance and skills will be provided in section 2 for each model as explained in the previous response.

*Section 2.2- same comment as above. We need to understand how EROS has been calibrated, and its resulting skill.*

We agree. We believe the response to the first comment will address this one.

*Section 2.3- Why not use a globally consistent forcing? I worry that this dataset is fine, but that we can't repeat these methods/data in other basins as the forcing is unique to France.*

In fact, T-NET can be applied to the any region where the input data including streamflow data (even supplied by other type of hydrological model rather than EROS model) are available. However, some modifications would be required according to the spatial and temporal resolution of input data. At first, it was decided to developed T-NET model over the Loire River basin since it encompasses an area with starkly contrasting land use/land cover, and climatic conditions (Moatar and Dupont, 2016), providing an ideal heterogeneity in both hydrological and thermal regimes. Now, this model is in preparation to be used over another basin in France (Saone) while the hydrological and hydrological related variables are provided by another hydrological model with higher spatial resolution.

We will address the model application more clearly in section 2 of the revised version.

*Figure 1- Is much too hard to read. The text size is too small. The resolution is quite high, but the figure is not legible without zooming in quite a lot. Please remake with readable font sizes*

We agree. We will provide a figure with a larger font size.

*Figure 2- I am not sure what this adds? I would delete it.*

We agree and we put this Figure in SM.

*3.1- this is not the place for skill scores of EROS. your model is about temperature, not discharge, so this information belongs in methods, not results.*

We agree that model skills were not provided in the right place. As we are providing the reader with both discharge and temperature dataset, their skill scores will be provided separately in section 2 (please see the response to the first comment).

*L127- "pretty good" is unacceptable professional writing. The atuhors must at least quantify how good the preformance is using some standard metric (RMSE, NSE, KGE, etc.)*

We agree. We consider this comment in the new manuscript and we will provide RMSE and NSE as well as biases for each model (Section 2).

*Figure 3 and L110- what is the justification for the 'locally weighted smoothing'? Are you using these lines to justify your trends? you must use a statistical test for these trends- either an MK or an PWMK or other citable and defensible test.*

The MK test was already used to address trends in retrospective simulations (see Seyedhashemi et al., 2022). This line shows only graphically the direction of changes (increasing, decreasing or rather statble) from the past to future using local regression models (Cleveland et al., 1992).

*Caveats- this is an odd section to include, and these skill scores belong elswhere, in results.*

We agree. In this regard, we removed this section and they will be written in the model calibration and validation section (please see the response to the new manuscript).

---

## Author Comment (AC2)

We thank the reviewer for generally positive comments. The reviewer comments are in italics and our responses are in normal font, the proposed text additions and modifications are in bold.

Please note that in the following, "P", "L" and "SM" stand for page number, line number and Supplementary Materials, respectively.

**#Reviewer 2:**

*The paper is interesting and present significant effort – very detailed dataset of historical and future stream temperature along Loire River basin. I would expect that it can be published after revision. The main concern regarding various parts of the manuscript are presented below.*

1.  *Rate and heterogeneity of discharge change during the historical period considering size of the basin seem too strong to be attributed to climate change solely. Some additional information about human impact on the discharge may clarify it.*

    Thank you for arising this point. As mentioned in P2L49-50 of the current manuscript, the hydrological and thermal models used here do not consider the influence of water-abstractions and impoundments i.e. simulate natural hydrological and thermal regimes. Therefore, the observed changes can just be attributed to the climate change. The changes in discharge is due to the fact that this basin encompasses an area with starkly contrasting climatic conditions.

2.  *It is not always clear what lower and upper limit of the range mean – "Indeed, 3 %–83% stations (resp. 50 %–100 %) on small and medium (resp. large) rivers had a RMSE< 1°C across seasons (see their Figure S9, bottom panel)". Is it variation across seasons, sizes?*

    We agree. This is across seasons and size. This will be clarified in the new manuscript.

3.  *Does the model considering potential landscape change due to climate change? Are there some estimates how significant this impact can potentially be?*

    No changes in land cover/land use was considered for both the hydrological and thermal models. Moreover, as mentioned in P3L84 for future projections, both hydrological and thermal models are run under present land cover/land use while calibrated parameters of hydrological model are kept as for the retrospective simulation. However, we completely agree on the potential impacts of land use/land cover on thermal regimes. In this regrad, we are working on another paper in which the influence of different scenarios of changes in riparian vegetation and following consequences on thermal regimes will be assessed.

---

## Author Comment (AC3)

We thank the reviewer for generally positive comments. The reviewer comments are in italics and our responses are in normal font, the proposed text additions and modifications are in bold.

Please note that in the following, "P", "L" and "SM" stand for page number, line number and Supplementary Materials, respectively.

**#Reviewer 3:**

This paper reconstructed historical, and projected future discharge and water temperature data with high spatial resolution for 52,278 reaches over the Loire River basin. This dataset will be useful for further researches in this area, and the method for the dataset generation might be able to be applied in more regions in the future. Overall, this manuscript is reasonably organized and I think this manuscript is acceptable for publication with minor revision.

*For all figures, if you want to describe the subfigures, please numbering each subfigure. For example, using a), b), c). And then using the numbers/letters refer to subfigures, instead of using the words like "top", "left" to locate them.*

We agree. We labeled panels with brackets around letters being lower case in this new manuscript.

*Line 65-69: Do you have validation period, if yes, please specify it.*

We agree. We added the period of validation data for both hyrological (1963-2019) and thermal models (2010-2014) in the new manuscript. We also added more information on calibration and validation of each model in the new manuscript.

*Line 71: "(see http://www.drias-climat.fr/", the bracket not closed.*

Thank you !

*Line 103: "the Nash-Sutcliffe efficiency of simulated daily Q is > 0.7 for Q, ln(Q), and √Q", do you mean the NSEs of Q, ln(Q), and √Q are all >0.7? Since you already have the NSE of daily Q, why the NSE of ln(Q), and √Q still should be considered?*

Yes, the NSEs of all Q, ln(Q), and √Q is > 0.7.

Maximizing the NSE criteria on the untransformed streamflow (*ln(Q), and √Q*) favors the goodness of fit of the hydrograph for high flows. Using the NSE criterion on the square roots

of the flows provides an estimate of model performance without favoring either high or low flows *Line 105: What would be the possible reason for the underestimation of Q in winter and spring?*

Using the square root of NSE does not favor neither high flows nor low flows. The underestimation of Q cannot be explained by this calibration choice.